# Opportunistic Screening for Osteoporosis by CT as Compared with DXA

**DOI:** 10.3390/diagnostics14242846

**Published:** 2024-12-18

**Authors:** Molaya Chaisen, Chanika Sritara, Niyata Chitrapazt, Chaiyawat Suppasilp, Wichana Chamroonrat, Sasivimol Promma, Arpakorn Kositwattanarerk, Chaninart Sakulpisuti, Kanungnij Thamnirat

**Affiliations:** 1Division of Nuclear Medicine, Department of Diagnostic and Therapeutic Radiology, Faculty of Medicine Ramathibodi Hospital, Mahidol University, Bangkok 10400, Thailand; molayamean@gmail.com (M.C.); chanika.sri@mahidol.edu (C.S.); wichana.cha@mahidol.edu (W.C.); sasipromma@gmail.com (S.P.); arpakorn.kos@mahidol.edu (A.K.); chaninart.sak@mahidol.ac.th (C.S.); 2Division of Nuclear Medicine, Department of Radiology, Maharaj Nakorn Si Thammarat Hospital, Naimueng, Mueng, Nakorn Si Thammarat 80000, Thailand; 3Division of Diagnostic Radiology, Department of Diagnostic and Therapeutic Radiology, Faculty of Medicine Ramathibodi Hospital, Mahidol University, Bangkok 10400, Thailand; niyata.chr@mahidol.ac.th; 4Department of Clinical Epidemiology and Biostatistics, Faculty of Medicine Ramathibodi Hospital, Mahidol University, Bangkok 10400, Thailand; chaiyawat.sup@mahidol.edu

**Keywords:** osteoporosis, non-contrast computed tomography, dual-energy X-ray absorptiometry, Hounsfield units, bone mineral density

## Abstract

Background: Osteoporosis is commonly evaluated using dual-energy X-ray absorptiometry (DXA) for bone mineral density (BMD). Non-contrast computed tomography (CT) scans provide an alternative for opportunistic osteoporosis assessment. This study aimed to evaluate screening thresholds for osteoporosis based on CT attenuation values in Hounsfield units (HU) of L1–L4 vertebrae from CT scans of the abdominal region, compared to DXA assessments of the lumbar spine and hips. Methods: Conducted retrospectively over approximately two years, the analysis included 109 patients who had both CT and DXA scans within 12 months, excluding those with metal artifacts affecting the vertebrae. CT attenuation values in the trabecular region of the vertebrae were measured and compared among three groups based on the lowest T-score from DXA. Results: In a predominantly female cohort (mean age 66.3 years), the lowest CT attenuation values for L1–L4 vertebrae showed a moderate correlation with the lowest T-score, with a Pearson correlation coefficient of 0.542 (95% CI: 0.388, 0.667). A HU threshold of ≤142 at the L1 vertebra showed 91.9% sensitivity and 48.4% specificity, while a threshold of ≤160 HU showed 97.3% sensitivity and 31.3% specificity for screening osteoporosis. Conclusions: This study supports the use of non-contrast CT with these HU thresholds as an opportunistic tool for osteoporosis assessment.

## 1. Introduction

Osteoporosis is a major public health concern characterized by reduced bone mass and deterioration of bone tissue, leading to increased fracture risk. The condition is often referred to as a “silent disease” due to its asymptomatic nature until a fracture occurs. Thus, effective screening methods are essential for identifying individuals at high risk and implementing preventive measures.

Dual-energy X-ray absorptiometry (DXA) is the gold standard for bone mineral density (BMD) measurement and osteoporosis diagnosis due to its precision and low radiation exposure [1,2,3,4]. However, it is less available in hospitals compared to computed tomography (CT). Quantitative CT (QCT) offers a direct method to assess BMD using CT attenuation [5,6,7,8,9,10]. Many studies suggest that QCT may be more sensitive than DXA for detecting osteoporosis in postmenopausal women, particularly by avoiding the overestimation of BMD associated with factors like spinal degeneration and abdominal aortic calcification. This enhanced sensitivity supports the consideration of QCT as a valuable tool in osteoporosis screening, though it requires calibration for accuracy. In contrast, standard CT scans can also provide BMD assessments in Hounsfield units (HU) without calibration, ensuring consistent interpretation across machines [11,12,13,14,15,16,17,18,19,20,21,22,23,24]. Thus, opportunistic screening for osteoporosis by assessing bone density can be considered along with routine CT scans. Thereby, this approach may not only diminish limited resource consumption including time, workloads, and costs, but also minimizes additional radiation exposure to the patients.

The International Society for Clinical Densitometry (ISCD) recommends evaluating BMD from CT by selecting two to three lumbar vertebrae levels from T12 to L4 [3,4]. However, only the first lumbar vertebra (L1) is typically used due to its ease of identification and its common inclusion in CT images of the thorax or abdomen, which enhances the opportunity for bone density assessment in routine clinical settings.

Previous studies have reported diagnostic performances of CT attenuation using various cutoffs [11,12,13,14,15,16,17,18,19,20,21,22,23,24]. Many pivotal studies, primarily involving large cohorts of Caucasian patients, have supported the use of CT for opportunistic osteoporosis screening. However, there is a notable gap in research focusing on Asian populations, which may limit the applicability of these findings. Variations in screening protocols, specific machines, and demographic factors can significantly influence these results.

The ISCD 2019 guidelines [4] emphasize the importance of opportunistic CT-based attenuation using Hounsfield Units (HU) for assessing bone health. Specifically, values such as L1 HU < 100 indicate a higher likelihood of osteoporosis, while L1 HU > 150 suggests normal bone density. This approach supports clinical decisions regarding further bone health assessments, particularly in the context of elective orthopedic and spine surgery, where patients may present with various risk factors for impaired bone health.

This research aims to evaluate appropriate screening thresholds for identifying high-risk individuals for osteoporosis using CT, compared to standard osteoporosis diagnosis from DXA measurements, and in relation to other established cutoffs among the Thai population, as the pioneering study of its kind in Southeast Asia.

## 2. Materials and Methods

This retrospective study adopts a cross-sectional analytic research design aimed at evaluating the association between CT attenuation values and bone mineral density (BMD) in adults. The primary focus is on individuals who have undergone both abdominal CT scans and dual-energy X-ray absorptiometry (DXA) bone density measurements within a defined timeframe.

### 2.1. Study Population

The target population comprises postmenopausal women and men aged 50 and older. To qualify for inclusion, participants must have received a CT scan of the abdomen or a CT scan of the kidneys, ureters, and bladder (KUB) that included coverage of the lumbar spine (L1–L4), in addition to having a DXA scan performed within a 12-month period, regardless of the order of the tests. The study examines data collected from April 2016 to November 2018.

To maintain the validity of the findings, specific exclusion criteria were established. Participants with metallic implants located in the abdominal region or lower lumbar spine, as well as patients with a history of vertebroplasty or other spine procedures, were excluded, as these implants could significantly interfere with CT attenuation measurements, making it impossible to accurately assess any lumbar vertebra.

Data collection encompassed a wide range of variables. Key demographic information was gathered, including age, sex, weight, height, and body mass index (BMI). In addition to these measurements, medical histories were carefully reviewed to identify underlying conditions associated with bone loss, such as primary hyperparathyroidism, chronic renal failure, hyperthyroidism, vitamin D deficiency, and Cushing’s syndrome. The study also took into account medications that could contribute to bone density reduction, including steroids, aromatase inhibitors, and levothyroxine, as well as treatments aimed at reducing bone resorption, such as antiresorptive drugs.

### 2.2. Protocol of DXA and CT Scans

Bone mineral density (BMD) assessments were performed on patients using Hologic Discovery DXA scanners (Hologic, Marlborough, MA, USA). The specific model used was the Discovery A, with software version 13.4.2. Scans were performed in Fast Array mode. These assessments focused on key anatomical sites including the lumbar spine, femoral neck, and total hip. For diagnostic purposes, the analysis considered the lowest T-score obtained from these regions, adhering to the World Health Organization (WHO) classification criteria for osteoporosis as follows:T-score ≤ −2.5: Osteoporosis−2.5 < T-score < −1.0: Low bone massT-score ≥ −1.0: Normal

These three categories of patients, based on the lowest T-score, were used to correlate with the mean CT attenuation values.

CT scans of the abdomen or CT scans of the KUB system covering the lumbar spine (L1–L4) were conducted using a Philips IQon Spectral CT scanner (Philips, Naterland B.V., The Netherlands), operating with software version 9.0. and with all imaging performed with or without contrast. The series selected for CT attenuation measurement was strictly limited to non-contrast-enhanced CT images, as this approach eliminates the potential for contrast agents to interfere with the precise evaluation of bone mineral density.

### 2.3. CT Attenuation Measurement

CT attenuation values for each L1–L4 vertebrae were measured on axial images from non-contrast CT scans, using a bone window setting. The measurement was performed at the midpoint of each vertebra’s height, with the Region of Interest (ROI) placed specifically on the trabecular bone. The ROI was drawn as large as possible, avoiding areas with artifacts, degenerative changes, bone islands, or hemangioma, as shown in Figure 1. The results are reported in Hounsfield units (HU) [11].

If the patient had vertebral compression graded at Grade I or higher, based on the Semiquantitative Grading for Vertebral Fracture [25], that vertebral level was excluded from analysis, while the remaining levels were measured as usual.

CT attenuation values were analyzed as mean CT attenuation in Hounsfield units (HU) for each vertebra and area of measurement in mm^2^.

Each vertebra was measured twice by the same radiologist with at least a one-month interval to assess the reliability of the measurement. The PACS (Picture Archiving and Communication System) of the hospital was used to ensure consistency and standardization in the imaging process.

### 2.4. Statistical Analysis

The statistical analysis included several key components. Descriptive statistics were used to present continuous data as means with standard deviations (SD) or median with range, as appropriate, while categorical data were reported as frequencies and percentages. For assessing the reliability of CT attenuation measurement, the intraclass correlation coefficient (ICC) was applied using a two-way random effects model. One-way ANOVA assessed differences in mean CT attenuation values among groups. Post-hoc analyses for multiple comparisons with Bonferroni correction were performed if significant differences were found. Empirical ROC curve analysis was carried out with osteoporosis designated as a true positive, while low bone mass and normal bone mass were considered false positives. Thresholds for use as screening tools were selected to ensure a sensitivity of 90% or higher, especially at level L1 vertebra. The value with the highest Youden’s index, combined with a specificity of 90% or more, was analyzed for comparison with previous studies. Finally, the characteristic performance was assessed by evaluating sensitivity, specificity, accuracy, positive and negative predictive values, odds ratios, and likelihood ratios. These metrics were compared with threshold values from Pickhardt et al. [11]. All the analyses were done using STATA 12 with a level of significance set at 5%.

## 3. Results

### 3.1. Study Population Characteristics

After applying the inclusion and exclusion criteria, 109 patients who underwent both an abdominal CT and DXA within a 12-month period were identified. The median interval between the two examinations was 139 days (range: 2–364 days), with 57.8% of the patients having an interval greater than six months. The majority of these patients were female (107 patients, 98.2%), with a mean age of 66.3 years (SD 11.2).

Subjects were categorized into three groups based on the diagnosis derived from the lowest T-score on DXA (lumbar spine and hip). The majority of patients were diagnosed with low bone mass, followed by osteoporosis, and then normal bone mass, as detailed in Table 1. Although a trend suggested that patients in the osteoporosis group were older than those in the low bone mass and normal groups, the overall age distribution did not differ significantly among the groups. However, the mean BMI of patients in the osteoporosis and low bone mass groups was significantly lower compared to the normal group (*p* = 0.004 and *p* = 0.004, respectively).

### 3.2. CT Attenuation Values

The number of assessable vertebrae at each level from L1 to L4 varied among the 109 participants due to factors such as metallic artifacts and vertebral collapse. Some vertebrae were excluded from the analysis according to Genant’s semiquantitative method [25], which recommends excluding vertebrae with compression fractures at grade I or higher. This approach led to fewer vertebrae being included at each level than the total number of participants. Specifically, the analysis included 101 vertebrae at L1, 104 at L2, 100 at L3, and 105 at L4.

CT attenuation measurements demonstrated excellent reliability, with an ICC exceeding 0.96 for all vertebrae and a mean difference of approximately 1–4 HU, despite lower ICC values and greater differences when considering the measurement areas, as detailed in Table 2.

The lowest CT attenuation values for L1–L4 showed a moderate correlation with the lowest T-score, with a Pearson correlation coefficient of 0.542 (95% CI: 0.388, 0.667). When considering DXA diagnostic groups, CT attenuation values revealed significant differences across groups and their post-hoc comparisons (*p*-value < 0.05). The mean CT attenuation values were highest in the normal bone mass group and lowest in the osteoporosis group, illustrating a trend in which lower attenuation corresponds with lower BMD, as shown in Figure 2 and Appendix A, Table A1.

The analysis categorized measurement areas into three subgroups based on the lowest T-score from the lumbar spine, femoral neck, and total hip. No significant differences were found among the osteoporosis, low bone mass, and normal bone mass groups for L1–L4 (Appendix A, Table A2).

Figure 3 illustrates distinct differences in mean CT attenuation values of patients in different groups: panel (a) depicts a patient classified in the normal bone mass group, while panel (b) shows a patient in the osteoporosis group.

### 3.3. Screening Value Analysis

The mean CT attenuation values for each lumbar vertebra (L1–L4) were analyzed to determine appropriate screening thresholds that could distinguish osteoporosis from low bone mass and normal bone mass. Screening thresholds were selected to achieve a sensitivity of 90% or higher across all vertebrae. Additional analyses were performed specifically for L1, focusing on identifying CT attenuation values that would meet a specificity of 90% or higher (≤94 HU) and the highest Youden’s index (≤128 HU). The study also evaluated the cutoff previously reported by Pickhardt et al. [11]. These threshold analyses are detailed in Table 3, providing an in-depth look at diagnostic performance for screening.

## 4. Discussion

This study aimed to identify appropriate screening values to differentiate individuals at high risk for osteoporosis from those at lower risk and to subsequently confirm high-risk cases using standard DXA measurements. We measured and averaged CT attenuation values at the trabecular bone of L1–L4 vertebrae and categorized patients into three groups based on the diagnosis derived from the lowest T-score obtained from DXA (lumbar spine and hip). Our findings indicate that the mean CT attenuation in individuals diagnosed with osteoporosis was significantly lower than that in individuals with low bone mass and normal bone mass. This suggests that CT attenuation values align with bone mass measurements obtained through standard methods, with lower CT attenuation reflecting lower bone mass.

Diagnostic criteria for osteoporosis typically involve the lowest T-score from either the lumbar spine, femoral neck, or total hip. Consequently, we used average CT attenuation from the L1–L4 vertebrae, grouped by these criteria, to determine screening values. Our results showed that sensitivity for detecting osteoporosis exceeded 90% at L1–L4 for specific CT attenuation thresholds: ≤142 HU at L1 (91.9%), ≤138 HU at L2 (91.7%), ≤128 HU at L3 (91.2%), and ≤133 HU at L4 (92.1%). Specificity for these thresholds ranged from 38.3% to 48.4%. Consistent with the 2015 ISCD Official Positions-Part III, which recommends assessing the lumbar spine from T12–L4, the study emphasizes the L1 vertebra due to its ease of identification and frequent inclusion in chest or abdominal CT scans, enhancing osteoporosis screening opportunities. L2, L3, and L4 vertebrae may be somewhat comparable to L1 in their screening efficacy; thus, if L1 is unavailable, using L2, L3, or L4 could still provide some valuable diagnostic information.

When comparing our findings with those of [11,13,17], our results indicate that sensitivity for detecting osteoporosis at a threshold of ≤142 HU at L1 exceeded 90%, while at ≤160 HU at L1, it was higher at 97.3%. This aligns with the findings of previous studies, which reported a sensitivity of more than 90% at a threshold of ≤160 HU at L1. The comparison of attenuation thresholds for osteoporosis detection of L1 vertebra are summarized in Table 4.

The discrepancies in results across studies can be attributed to differences in ethnicity, sample size, equipment used, measurement methods, and the interval between CT and DXA scans. These variables significantly influence the outcomes of osteoporosis screening and diagnosis. One major factor is ethnicity, as variations in bone density and structure can occur among different racial and ethnic groups. This diversity can lead to differing baseline values for CT attenuation and T-scores, ultimately affecting diagnostic accuracy.

In addition to these factors, our results demonstrated a significant relationship between BMI and bone mass. Specifically, patients in the osteoporosis and low bone mass groups had significantly lower mean BMIs compared to those in the normal bone mass group (*p* = 0.004 and *p* = 0.004, respectively). This finding correlates with the WHO [3] and the National Osteoporosis Foundation (NOF) [26], both of which suggest that higher BMI, particularly in overweight and obese categories, is often linked to greater bone mineral density (BMD). This is likely due to the increased mechanical loading and pressure on bones, which can stimulate bone formation and contribute to higher BMD in individuals with higher BMI.

One notable limitation of this study is the small sample size, which may restrict the generalizability of our findings. Larger studies with more diverse cohorts are necessary to validate our findings and assess their applicability to different demographic groups. Additionally, larger sample sizes would help improve the precision of the estimates and provide a more comprehensive understanding of the relationship between CT attenuation and T-scores across various populations.

The type of equipment used also contributes to variability in the results. Different CT and DXA machines may have distinct calibration settings, imaging protocols, and sensitivity levels, all of which can impact the measurements obtained. Furthermore, the methodologies employed for measuring CT attenuation can vary, leading to inconsistent findings. Moreover, the interval between CT and DXA scans is crucial. Most studies recommend a time frame of no more than 6 months between the two assessments to ensure that changes in bone density are accurately captured. However, this study opted for a 12-month interval due to the limited number of patients who underwent both tests within a 6-month period. This longer interval may introduce additional variability, as changes in bone density can occur over time, particularly in individuals at higher risk for osteoporosis.

There are other considerations in this study. We acknowledge that fragility fractures are fractures that occur from minimal trauma, typically resulting from a fall from standing height or less and were not incorporated into the analysis presented here. While fragility fractures were considered in refining the classification [27], this study focused on grouping patients solely according to the lowest T-score criteria. Therefore, patients with low bone mass or normal BMD but with fragility fractures may have been misclassified as non-osteoporotic. This limitation is acknowledged, and future studies will explore the potential impact of including fragility fractures on the CT attenuation values and the practical implications of such a reclassification.

In terms of application, the choice of CT attenuation values for osteoporosis screening depends on the examination’s purpose. For screening, a threshold with high sensitivity is preferred. This study identified a threshold of ≤142 HU at L1 or ≤160 HU at L1, both of which offer high sensitivity. The ≤160 HU threshold has been confirmed in various previous studies, demonstrating its ability to detect osteoporosis with high sensitivity. Using a high-sensitivity threshold reduces the likelihood of missing osteoporosis in individuals with higher HU values, while those with lower HU values are referred for DXA confirmation. This approach enhances the detection of osteoporosis cases, allowing for timely treatment planning before fractures occur, thereby improving patient quality of life.

In personal practice, combining DXA with CT attenuation values provides several advantages beyond screening for osteoporosis. This approach enhances the overall assessment of bone health by allowing for a more detailed evaluation of trabecular bone quality, which is crucial in identifying patients at risk for fractures, especially in cases where BMD by DXA may be inaccurate due to factors such as severe bone spur formation or sclerosis, aortic calcifications, and obesity. This is particularly important for patients at high risk for osteoporosis, such as those with low trabecular bone scores or a history of low-trauma fractures. By incorporating CT attenuation values, clinicians can obtain a more accurate depiction of bone health, ultimately guiding better clinical decisions.

The effectiveness of using CT for opportunistic screening has been demonstrated in several studies [11,12,13,14,15,16,17,18,19,20,21,22,23,24]. Additionally, systematic reviews and meta-analyses confirm that CT-based methods can provide accurate assessments comparable to DXA, potentially enhancing screening programs.

## 5. Conclusions

This research demonstrates that abdominal CT scans, conducted for various clinical indications, can serve as a valuable screening tool for identifying individuals at high risk for osteoporosis, provided that appropriate screening values are utilized. This not only aids in efficient patient management but also contributes to overall healthcare resource optimization by reducing the need for additional radiation exposure and repeat imaging. This study identified a threshold of ≤142 HU at L1 and ≤160 HU at L1, both of which offer high sensitivity and have been confirmed in various studies. Notably, the threshold of ≤160 HU at L1 provides a higher sensitivity compared to ≤142 HU. These thresholds align with previously established values while offering a practical solution for utilizing existing CT scan data.

In conclusion, the findings emphasize the feasibility of integrating CT attenuation measurements into osteoporosis screening protocols, highlighting the importance of continuous refinement of these parameters to enhance diagnostic accuracy and patient care. Future investigations should further explore the longitudinal impact of using CT attenuation as a routine screening method in diverse populations, ultimately contributing to improved osteoporosis management and patient outcomes.

## Figures and Tables

**Figure 1 diagnostics-14-02846-f001:**
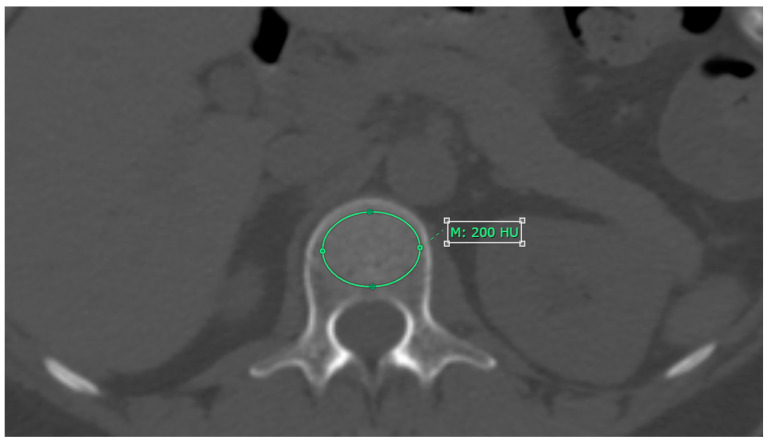
The method for measuring CT attenuation values in cross-sectional images involves placing the region of interest (ROI) specifically on the trabecular bone area. In this approach, M represents the mean CT attenuation value.

**Figure 2 diagnostics-14-02846-f002:**
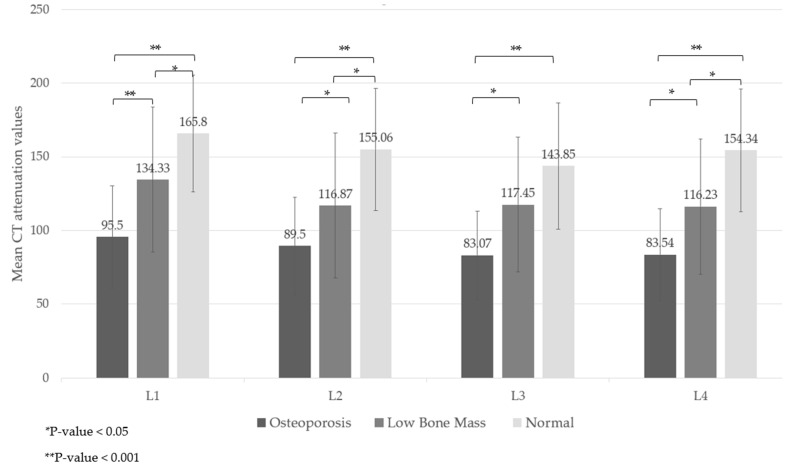
Mean CT attenuation values of L1–L4 vertebrae categorized by the DXA diagnosis by the lowest T-score.

**Figure 3 diagnostics-14-02846-f003:**
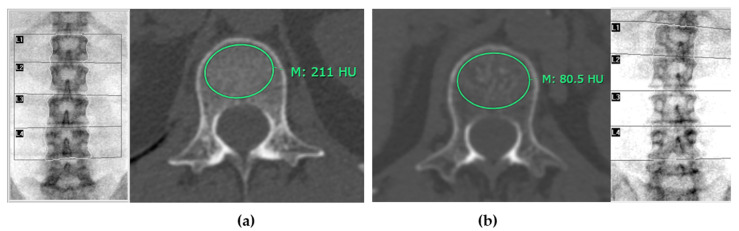
Mean CT attenuation of patients in different groups. Panel (**a**) illustrates a patient classified in the normal bone mass group based on the lowest T-score from DXA, exhibiting a mean CT attenuation of approximately 211 Hounsfield Units (HU). In contrast, panel (**b**) depicts a patient in the osteoporosis group, characterized by a significantly lower mean CT attenuation of 80.5 HU. This comparison underscores the distinct differences in bone density as measured by CT attenuation values across these classifications.

**Table 1 diagnostics-14-02846-t001:** Characteristics by diagnosis based on the lowest T-score.

Characteristic	Total n = 109 (100%)	Diagnosis Based on the Lowest T-Score			
		Osteoporosis n = 39 (35.8%)	Low Bone Mass n = 51 (46.8%)	Normal n = 19 (17.4%)	*p*-Value
Age (years), mean ± SD	66.32 ± 11.24	67.64 ± 10.62	65.67 ± 11.27	65.37 ± 12.86	0.660
Gender					0.662
-Female	107 (98.2)	39 (36.4)	49 (45.8)	19 (17.8)	
-Male	2 (1.8)	-	2 (100)	-	
BMI (kg/m^2^), mean ± SD	24.2 ± 4.4	23.4 ± 4.5	23.6 ± 3.8	27.3 ± 4.8	0.003
Underlying Conditions Causing Bone Loss	21 (19.3)	8 (38.1)	10 (47.6)	3 (14.3)	1.000
Medications Causing Bone Loss	23 (21.1)	8 (34.8)	12 (52.2)	3 (13.0)	0.863
Medications Reducing Bone Resorption	67 (61.5)	25 (37.3)	31 (46.3)	11 (16.4)	0.893
Days between CT and DXA (days), median (range)	139 (2–364)	143 (15–332)	139 (2–364)	112 (14–293)	0.443 ^#^ 0.498
-6 months or less	46 (42.2)	16 (34.8)	24 (52.2)	6 (13.0)	
-6–12 months	63 (57.8)	23 (36.5)	27 (42.9)	13 (20.6)	

^#^ Kruskal-Wallis test.

**Table 2 diagnostics-14-02846-t002:** Reliability of CT attenuation measurement of L1–L4 vertebrae.

Measures	Difference (95%CI)	ICC (95%CI)	*p*-Value
HU			
L1	1.307 (−0.327, 2.942)	0.986 (0.979, 0.990)	<0.001
L2	0.456 (−0.620, 1.531)	0.994 (0.990, 0.996)	<0.001
L3	0.878 (−0.789, 2.544)	0.983 (0.975, 0.989)	<0.001
L4	1.425 (−0.936, 3.787)	0.966 (0.951, 0.977)	<0.001
Area			
L1	17.403 (4.901, 29.904)	0.755 (0.649, 0.83)	<0.001
L2	19.143 (8.626, 29.661)	0.870 (0.798, 0.916)	<0.001
L3	40.362 (22.805, 57.919)	0.738 (0.574, 0.834)	<0.001
L4	55.140 (36.108, 74.171)	0.714 (0.486, 0.831)	<0.001

**Table 3 diagnostics-14-02846-t003:** CT attenuation values of L1–L4 vertebrae for screening individuals at risk of osteoporosis.

Level	HU	Sensitivity (95% CI)	Specificity (95% CI)	Accuracy (95% CI)	LR+ (95% CI)	LR− (95% CI)	PPV (95% CI)	NPV (95% CI)
L1	≤94	48.7 (31.9–65.6)	90.6 (80.7–96.5)	75.2 (65.7–83.3)	5.2 (2.3–11.9)	0.6 (0.4–0.8)	75 (56.7–87.3)	75.3 (68.8–80.8)
	≤128	86.1 (70.5–95.3)	62.5 (49.5–74.3)	71.0 (61.1–79.6)	2.3 (1.6–3.2)	0.2 (0.1–0.5)	56.4 (47.8–64.5)	88.9 (77.6–94.9)
	≤142	91.9 (78.1–98.3)	48.4 (35.8–61.3)	64.41 (54.2–73.6)	1.8 (1.4–2.3)	0.2 (0.1–0.5)	50.8 (44.4–57.1)	91.2 (77.2–96.9)
	≤160	97.3 (85.8–99.9)	31.3 (20.2–44.1)	55.5 (45.2–65.3)	1.4 (1.2–1.7)	0.1 (0.01–0.6)	45 (40.1–49.3)	95.2 (73.7–99.3)
L2	≤138	91.7 (77.5–98.3)	38.3 (26.7–50.8)	56.7 (46.7–66.4)	1.5 (1.2–1.8)	0.2 (0.07–0.7)	44.0 (38.9–49.3)	89.7 (73.8–96.4)
L3	≤128	91.2 (76.3–98.1)	43.9 (31.7–56.7)	60.0 (49.7–69.7)	1.6 (1.3–2.1)	0.2 (0.07–0.6)	45.6 (39.8–51.5)	90.6 (76.0–96.7)
L4	≤133	92.1 (78.6–98.3)	40.3 (28.5–53.0)	59.1 (49.0–68.6)	1.5 (1.2–1.9)	0.2 (0.06–0.6)	46.7 (41.3–52.1)	90.0 (74.5–96.5)

LR+ = Positive Likelihood Ratio, LR− = Negative Likelihood Ratio, PPV = Positive Predictive Value, NPV = Negative Predictive Value.

**Table 4 diagnostics-14-02846-t004:** Comparison of attenuation thresholds for osteoporosis detection of L1 vertebra.

Study	Threshold (HU)	
	Sensitivity ≥ 90%	Specificity ≥ 90%
This Study	≤142 and ≤160	≤94
[11]	≤160	≤110
[13]	≤160	≤73
[17]	≤160	≤80

## Data Availability

The data presented in this study are available on request from the corresponding author.

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
