# Peer review of "Opportunistic Screening for Osteoporosis by CT as Compared with DXA"

_diagnostics, 2024, doi:10.3390/diagnostics14242846_

Round 1
Reviewer 1 Report
Comments and Suggestions for Authors
The study evaluates the assessment of bone mineral density (BMD) using attenuation values of lumbar vertebral bodies on body CT, compared with DXA, in the Thai population. The cut-off value for sensitivity and specificity of the measured HU is discussed.
Specific Comments:
Abstract:
1. L. 28: Merely stating that significant correlations were found between HU values and the lowest T-score is insufficient. Please include numerical values. Additionally, the percentages of the three groups mentioned in the subsequent sentence are unnecessary.
Keywords:
OK.
Introduction:
2. L. 65–80: The results of previous studies are repeated in the Discussion. Consider shortening this paragraph to avoid redundancy.
3. L. 86: Please insert a reference.
Materials and Methods:
4. L. 110: Were any patients with a history of vertebroplasty or other spine procedures included or excluded from the study? Please clarify.
5. L. 121: Define "fragility fractures."
6. L. 133: The incorporation of the history of fragility fractures into the classification criteria alongside the T-score is unclear. Please explain and reference the relevant WHO criteria. Additionally, clarify how this reclassification impacts the Results, including Figure 2.
7. L. 150: Insert a reference for SGVF.
8. L. 157: How many radiologists performed the ROI selection? Was the same or different radiologists involved in performing the selections one month apart?
Results:
9. L. 221: Insert a reference.
10. L. 242–252: The paragraph discussing the measurement area for CT attenuation across different lumbar spine levels appears inconsistent. Initially, it is suggested that this factor may impact measurement accuracy and reliability. However, the subsequent analysis reports no significant influence on bone mass categorization, which is reiterated in the Discussion. Finally, the need to consider the measurement area is emphasized again. Please clarify.
11. L. 277: In Table 3, explain the abbreviations LR+, LR-, PPV, and NPV in the table legend.
Discussion:
12. How do the authors explain the significantly higher BMI in patients with normal BMD?
13. L. 297: Consider presenting the comparison of this study with the findings of Pickhardt, Alacreu, Buckens, and Li in a table format for clarity.
14. L. 324: The statement about the role of sample size is too general. If it pertains to this study, describe it more specifically.
15. How do the authors explain the decreasing specificity with increasing HU thresholds for osteoporosis?
16. L. 339: Again, the mention of fragility fractures is confusing. While these fractures are included in the study (Materials and Methods, Figure 2), the Discussion states that the analysis does not account for patients with fragility fractures. Please clarify.
Figures and Tables: discussed above.
Reviewer 2 Report
Comments and Suggestions for Authors
Thank you for inviting me to review this manuscript. Opportunistic osteoporosis screening is important because it reduces the number of times patients undergo tests involving ionizing radiation. My major contention about this paper is how the effective cutoffs were selected. Selecting a higher cut-off will always result in higher sensitivity at the expense of specificity. Since the authors already conducted ROC analysis and established new cutoffs (128), why do they still select a higher cutoff (160)? This would require some explanation. The implications of selecting such a high cutoffs should also be explained.
Minor comments:
Please demonstrate sample size calculation since the numbers of subjects is moderate.
Line 136: For every instrument and software used, please clearly indicate the mode, version, manufacturer, city and country of the company.
In Table 1, what test did you use for the comparison between categorical data? Especially data involving gender with many cells having <5 number, what test you used for comparison?
For Table 1, what are the underlying conditions that predispose patients to osteoporosis?
Last row for Table 1, please align the p-value properly.
For all ANOVA results (Fig 3, Table A1 and A2), please indicate the pairwise comparison.
Round 2
Reviewer 1 Report
Comments and Suggestions for Authors
Comments and R 12: The explanation is proper however it should be moved to Discussion.
